# LANGUAGE-ASSISTED SUPER-RESOLUTION FROM REAL-WORLD LOW-RESOLUTION PATCHES

## ABSTRACT

Single image super-resolution (SISR) aims to reconstruct high-resolution (HR) images from low-resolution (LR) inputs. Training SR models typically requires paired HR–LR data, which is difficult to obtain in reality. As a result, most methods synthesize LR images by artificially degrading HR images with handcrafted kernels or camera ISP adjustments. However, these synthetic degradations fail to capture the complexity of real LR images, leading to poor generalization in practice. To address this, we observe that even within a single high-quality image, regions at different depths exhibit varying resolutions—where distant regions act as LR patches and closer ones as HR patches. This allows the extraction of real, degradation-induced LR patches from real images. Since these LR patches lack paired HR counterparts, we propose LA-SR (Language Assistant for SR), a novel framework for unpaired SR. The key idea of LA-SR is to redefine unpaired SR in the *language space*, using vision-language models to bridge the LR–HR gap. LA-SR projects images into a semantic-rich space representing both content and quality, and applies two language-guided losses: linguistic-content loss to preserve semantic fidelity, and linguistic-quality loss to enhance perceptual realism. With this alignment, LA-SR effectively super-resolves real LR inputs, producing realistic outputs that overcome the limitations of synthetic-data-trained methods.

## 1 INTRODUCTION

Despite advancements in camera technology, the quality of images may still fall short of human expectations, often requiring zooming to reveal finer details. To address this, early super-resolution (SISR or SR) methods Lim et al. (2017b); Ledig et al. (2017); Zhang et al. (2018c;b); Fan et al. (2020); Park et al. (2023) have focused on recovering high-resolution (HR) images from degraded low-resolution (LR) counterparts, where LR images were generated by degrading HR images Zhang et al. (2021); Wang et al. (2021b); Joze et al. (2020); Yue et al. (2022); Cai et al. (2019).

Specifically, early LR images Timofte et al. (2017) are constructed by downsampling (*e.g.*, bicubic and bilinear) HR images, failing to capture real-world degradations. To overcome this, later methods have explored two main strategies for constructing LR images that more accurately reflect real-world degradations. One method Zhang et al. (2021); Wang et al. (2021b) synthetically simulates image signal processor (ISP) pipelines, incorporating various degradation processes (*e.g.*, noise, blur, and compression). However, they often fail to capture the full complexity of real-world degradations. The other methods utilize specialized camera systems with beam splitters Joze et al. (2020); Yue et al. (2022) or multiple focal lengths Cai et al. (2019) to capture realistic LR images. However, they require sophisticated hardware and still rely on a specific camera model, restricting their ability to generalize to complex real-world degradations.

On this basis, we propose a novel SR framework, the Language Assistant for SR (LA-SR), to address this. Inspired by DGDML-SR Cheng et al. (2020), we observe that even high-quality images often contain both real-LR and HR regions due to varying subject-to-camera distances, as shown in Figure 1 (*e.g.*, distant grass and close tiger). This depth variation causes different regions within the same image to exhibit distinct levels of detail, effectively serving as unpaired LR-HR patches. However, the absence of explicit LR-HR pairs makes direct supervised training difficult. Although DGDML-SR Cheng et al. (2020) addresses this issue by identifying similar content patterns within the same image to construct LR-HR pairs, this assumption rarely holds in real-world scenarios,

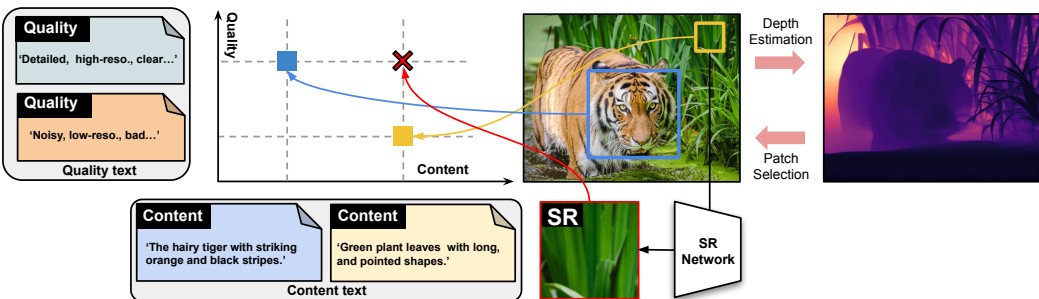

Figure 1: **Overview of LA-SR**. Based on the subject's distance from the camera, an image can contain both LR and HR regions. Based on this, we use depth information for the distance, segmenting real-LR and HR patches. Then, from extracted **LR** and **HR** patches, each is encoded to strongly correlate with its corresponding content texts, with **LR** patches aligned to low-quality texts and **HR** patches to high-quality texts. Then, the SR network is trained to produce SR images that align with the input's content and high-quality texts, ensuring both content preservation and high detail.

limiting the applicability of their approach. Instead, to address this limitation, we leverage powerful vision-language models Li et al. (2022); Radford et al. (2021); Touvron et al. (2023); Li et al. (2024), which are capable of understanding both the semantic content and perceptual quality of images, enabling effective supervision without requiring explicit LR-HR pairings.

To elaborate, we utilize the correlation between language and image in our LA-SR framework, as shown in Figure 1. Here, considering content and quality as crucial factors in image restoration Saha et al. (2023); Zhao et al. (2023), we present two loss terms: *linguistic-content loss* to preserve the content of the input images in the SR images and *linguistic-quality loss* to ensure the high quality of the SR images. Specifically, using a depth map obtained from a pre-trained model Gui et al. (2025), we extracted LR-HR patches from high-quality images: large HR patches (*e.g.*, **blue patch** in Figure 1) from near depths and small LR patches (*e.g.*, **yellow patch** in Figure 1) from distant depths. Then, we prepare two types of textual descriptions: content texts, derived from each patch using a pre-trained image-to-text model Li et al. (2024); Touvron et al. (2023); Radford et al. (2021), and pre-defined quality texts, categorizing patches as either high or low quality. Assuming distant LR patches contain fewer details than closer ones, we use contrastive learning to encode LR patches with strong correlations to their respective content and low-quality texts, while maintaining weak correlations with other texts (and vice versa for HR patches) (*e.g.*, ■ and ■ in Figure 1). Finally, we train our LA-SR framework to produce SR images (*e.g.*, **red patch** in Figure 1) that align closely with their corresponding content and high-quality texts (*e.g.*, ✖ in Figure 1), thereby maintaining the content of input patches and preserving the details of HR patches.

LA-SR effectively produces realistic results from diverse real-world inputs by training with real LR patches and incorporating our language-based loss terms: linguistic-content and linguistic-quality losses. Instead of relying on previously used synthetically degraded LR benchmarks Timofte et al. (2017); Zhang et al. (2021); Wang et al. (2021b), we show the efficacy of LA-SR on natural images, including various benchmark datasets. We summarize our contributions as follows:

- **Unpaired Real-World SR Framework:** We propose LA-SR, a novel framework that learns from real LR patches extracted from high-quality images using depth information, eliminating the need for synthetic degradations.

- **Language-Space Alignment with Guided Losses:** We formulate unpaired SR as a language-space alignment problem using a pretrained vision-language model, and design content and quality losses to keep semantics and enhance realism, enabling high-quality SR.

- **Superior SR Performance:** Our LA-SR delivers superior SR performance across various benchmarks, excelling in both perceptual metrics and visual quality, demonstrating strong generalization to real-world LR inputs.

## 2 RELATED WORKS

**Real-world Super-Resolution.** Since the early breakthrough of deep learning-based SR methods such as VDSR Kim et al. (2016), EDSR Lim et al. (2017b), and SRGAN Ledig et al. (2017), many subsequent SR methods Zhang et al. (2018b); Fan et al. (2020); Wang et al. (2022) have supervised their networks by minimizing distance-based losses (*e.g.*, pixel-wise and VGG) between super-resolved and high-resolution images. Although they achieve strong performance, they often struggle with real-world images due to reliance on LR images Timofte et al. (2017); Wang et al. (2018a); Lim et al. (2017a) generated through simple downsampling that fail to replicate real degradations. A straightforward solution is to create a dataset that better reflects real LR images. To do this, several methods Wang et al. (2021b); Zhang et al. (2021) apply degradation-based augmentations (*e.g.*, noise, resizing, and JPEG compression) but fail to capture real-world complexities. Others use specialized camera setups (*e.g.*, beam splitters Joze et al. (2020); Yue et al. (2022) or multiple focal lengths Cai et al. (2019)) to capture LR images with actual ISP degradations. However, they require advanced equipment and produce LR images that are specific to particular camera models, limiting generalizability.

**Self-supervised Super-Resolution.** Without requiring LR-HR pairs, several zero-shot methods Shocher et al. (2018); Soh et al. (2020) have proposed training strategies using only LR images. ZSSR Shocher et al. (2018) trains an image-specific CNN on a given LR image, while MZSR Soh et al. (2020) uses an initialization from a large-scale external dataset Timofte et al. (2017) to tailor the model to given LR images. However, they assume that LR images are degraded with a known degradation, which is often impractical. To tackle images with unknown degradations, subsequent methods use auxiliary reference images, either captured via dual-camera setups Wang et al. (2021a); Zhang et al. (2022) or retrieved using patch similarity Lu et al. (2021), and train SR networks in a self-supervised manner. However, obtaining reference images is challenging, and they still rely heavily on manually degraded LR images, which limits their applicability to real-world images. DGDML-SR Cheng et al. (2020) takes a different approach by using depth-based LR-HR patch and employing a cyclic GAN Zhu et al. (2017). However, due to the instability of the adversarial training, their method restricts the LR-HR patch pairs to regions with highly similar content (*e.g.*, identically shaped windows), which is an unrealistic constraint in most real-world scenarios.

**Language in image restoration.** Although early image restoration methods focused solely on image data, recent approaches recognize the strong correlation between image and language and use language to enhance restoration. Specifically, CoSeR Sun et al. (2024) utilizes cognitive embeddings with semantic and textural information for super-resolution. LLMRA Jin et al. (2024) leverages language priors based on user dialogue to restore various degraded images. Similarly, LM4LV Zheng et al. (2024) uses a frozen large-language model (LLM) to generate visual tokens, which are then decoded into restored images. These advances highlight the potential for further integration of language assistance into image restoration.

**Contrastive Language-Image training.** Image-based contrastive learning aims to learn discriminative representations to differentiate an image from others, proving effective across various tasks. Starting from simple classifications Chen et al. (2020) that map images to corresponding labels (*e.g.*, cat and dog), later works Radford et al. (2021); Yang et al. (2022); Yu et al. (2022) have adopted contrastive learning to capture the correlation between images and text descriptions. Specifically, they jointly train image and text encoders to predict correlated pairs from a batch of (image, text) data. They combine pairs of images and corresponding text descriptions in a shared embedding space while pushing apart pairs that do not match. In other words, instead of learning a one-to-one mapping between image and text, these approaches determine how multiple images are relatively correlated to each text.

## 3 PROPOSED METHOD

Figure 2 shows the overall pipeline of our LA-SR. We utilize an existing SR network Wang et al. (2018b) without modifications for our SR network, while introducing two novel loss terms: linguistic-content and linguistic-quality losses.

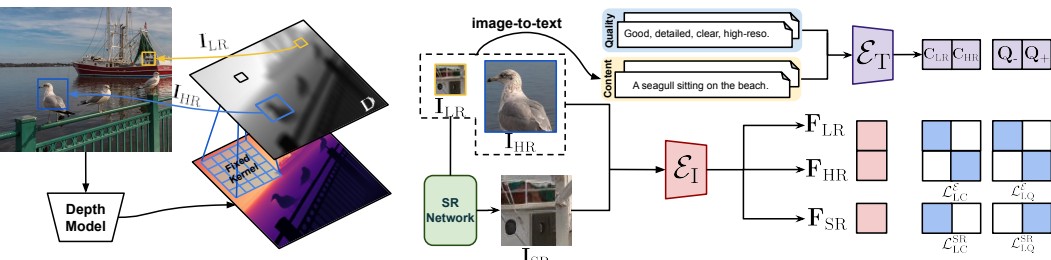

(a) LR-HR patches preparation      (b) Overview of training framework

Figure 2: **Overview of LA-SR.** (a) We extract $\mathbf{I}_{LR}$ and $\mathbf{I}_{HR}$ from images using estimated depth $\mathbf{D}$. (b) The linguistic-content loss $\mathcal{L}_{LC}$ classifies patches based on their content text features, training the SR network to produce outputs aligned with their corresponding content. Meanwhile, the linguistic-quality loss $\mathcal{L}_{LQ}$ distinguishes $\mathbf{I}_{LR}$ and $\mathbf{I}_{HR}$ patches based on quality, guiding the SR network to produce outputs $\mathbf{I}_{SR}$ aligning the $\mathbf{I}_{HR}$ distribution. The figure shows LA-SR with a batch size= 1.

## 3.1 PREPARING LR-HR PATCHES

We begin by extracting LR patch $\mathbf{I}_{LR}$ and HR patch $\mathbf{I}_{HR}$ from high-quality images. To achieve this, we first use a pre-trained depth estimation network Gui et al. (2025) to determine the relative distance of each pixel from the camera, selecting distant patches as $\mathbf{I}_{LR}$ and closer patches as $\mathbf{I}_{HR}$, as shown in Figure 2a. Here, unlike the previous depth-based approach Cheng et al. (2020), which constrains LR and HR patches to share similar textures (*i.e.*, patterns), our method imposes no such constraint, allowing LR-HR patches to be extracted from any image regardless of content similarity. To capture the depth of pixels within each patch, we apply a convolution with a large fixed kernel (*i.e.*, all values set to 1 and of size $\mathbb{R}^{1\times3\times29\times29}$), enabling depth estimation based on surrounding pixels, resulting in the final depth map $\mathbf{D}$. Then, using the given image and estimated depth map $\mathbf{D}$, we extract two types of patches: small patches from distant regions $\mathbf{I}_{LR} \in \mathbb{R}^{h\times w\times3}$ and large patches from closer regions $\mathbf{I}_{HR} \in \mathbb{R}^{H\times W\times3}$. Here, $(h, w)$ and $(H, W) = (sh, sw)$ are the (height, width) of each patch with $s > 1$ as a fixed scaling factor. Specifically, we select $\mathbf{I}_{LR}$ and $\mathbf{I}_{HR}$ from the depth map $\mathbf{D}$ using a top-$k$ algorithm, where the $k$ farthest pixels (*i.e.*, bright regions in Figure 2a) are assigned as centers of $\mathbf{I}_{LR}$ and the $k$ nearest pixels (*i.e.*, dark regions in Figure 2a) are selected as centers of $\mathbf{I}_{HR}$. During training, we super-resolve $\mathbf{I}_{LR}$ into a higher-resolution SR image $\mathbf{I}_{SR} \in \mathbb{R}^{H\times W\times3}$. Training supervises this process to ensure that $\mathbf{I}_{SR}$ retains the content of $\mathbf{I}_{LR}$ via linguistic-content loss while also adhering to the details of $\mathbf{I}_{HR}$ through linguistic-quality loss, as shown in Figure 2b.

## 3.2 ARCHITECTURE OF ENCODERS

LA-SR employs contrastive learning between images and text, using the image encoder $\mathcal{E}_I$ and text encoder $\mathcal{E}_T$ from the pre-trained CLIP model Radford et al. (2021), both designed with a Transformer Vaswani et al. (2017). Here, since the original CLIP model has a fixed input size (*e.g.*, $224 \times 224$), we follow the approach in Wang et al. (2023b) by removing the positional embedding from the image encoder $\mathcal{E}_I$, allowing it to accept variable input sizes.

## 3.3 LINGUISTIC-CONTENT LOSS

Previous SR methods Zhang et al. (2018c); Fan et al. (2020) directly compare $\mathbf{I}_{SR}$ and $\mathbf{I}_{HR}$ to preserve content, which is not applicable in our case due to unpaired LR-HR patches. To address this, we introduce a linguistic-content loss $\mathcal{L}_{LC}$ to ensure that $\mathbf{I}_{SR}$ retains the content of $\mathbf{I}_{LR}$. To achieve this, we first generate content text from $\mathbf{I}_{LR}$ and $\mathbf{I}_{HR}$ using an image-to-text model Li et al. (2022). Then, the text encoder $\mathcal{E}_T$ processes these content texts, producing $\mathbf{C}_{LR}$ from $\mathbf{I}_{LR}$ and $\mathbf{C}_{HR}$ from $\mathbf{I}_{HR}$. Similarly, image features $\mathbf{F}_{LR}$, $\mathbf{F}_{HR}$, and $\mathbf{F}_{SR}$ are extracted from $\mathbf{I}_{LR}$, $\mathbf{I}_{HR}$, and $\mathbf{I}_{SR}$, respectively, using the image encoder $\mathcal{E}_I$. Note that all the above features are $\in \mathbb{R}^{1\times c}$, where $c$ represents the channel dimension and 1 represents the batch size when the batch size is set to 1.

Afterward, we jointly train the encoders and the SR network using text and image features. For the encoders, the loss function $\mathcal{L}_{\text{LC}}^{\mathcal{E}}$ is optimized using a contrastive function as:

$$\text{Cont}(\mathbf{A}, \mathbf{B}) = -\frac{1}{*} \sum_{i=1}^{*} \left( \log \frac{e^{\cos(\mathbf{A}_i, \mathbf{B}_i)}}{\sum_{k=1[k \neq i]}^{*} e^{\cos(\mathbf{A}_i, \mathbf{B}_k)}} \right), \tag{1}$$

$$\mathcal{L}_{\text{LC}}^{\mathcal{E}} = \text{Cont}(\mathbf{F}_{\text{LR}} \parallel \mathbf{F}_{\text{HR}}, \mathbf{C}_{\text{LR}} \parallel \mathbf{C}_{\text{HR}}),$$

where $\mathbf{A}$ and $\mathbf{B}$ represent any features in $\mathbb{R}^{* \times c}$, and $\cos(\cdot, \cdot)$ denote cosine similarity, where $\cos(\mathbf{A}_i, \mathbf{B}_i) = \frac{\mathbf{A}_i \cdot \mathbf{B}_i^T}{\|\mathbf{A}_i\| \|\mathbf{B}_i\|}$. Moreover, $\parallel$ denotes concatenation along the batch dimension. The proposed $\mathcal{L}_{\text{LC}}^{\mathcal{E}}$ trains the encoders to align images with their corresponding content texts, while contrasting them with unrelated images and content texts.

For the SR network, we compute the loss function $\mathcal{L}_{\text{LC}}^{\text{SR}}$ to ensure that the SR image $\mathbf{I}_{\text{SR}}$ maintains strong correlation with the corresponding content text feature $\mathbf{C}_{\text{LR}}$ as the inputs $\mathbf{I}_{\text{LR}}$, ensuring both share the same content as:

$$\mathcal{L}_{\text{LC}}^{\text{SR}} = \text{Cont}(\mathbf{F}_{\text{SR}}, \mathbf{C}_{\text{LR}}). \tag{2}$$

Finally, the linguistic-content loss $\mathcal{L}_{\text{LC}}$ is defined as sum of Equations 1 and 2 ($\mathcal{L}_{\text{LC}} = \mathcal{L}_{\text{LC}}^{\mathcal{E}} + \mathcal{L}_{\text{LC}}^{\text{SR}}$).

### 3.4 LINGUISTIC-QUALITY LOSS

For the quality of $\mathbf{I}_{\text{SR}}$, we design linguistic-quality loss $\mathcal{L}_{\text{LQ}}$. Unlike the content, where $\mathbf{I}_{\text{SR}}$ and $\mathbf{I}_{\text{LR}}$ should share the same content, $\mathbf{I}_{\text{SR}}$ should contain more details than $\mathbf{I}_{\text{LR}}$, as it uses more pixels to represent the same content. Therefore, assuming that patches from closer distances $\mathbf{I}_{\text{HR}}$ contain more details than those from distant ones $\mathbf{I}_{\text{LR}}$, we train the encoders to distinguish between $\mathbf{I}_{\text{LR}}$ and $\mathbf{I}_{\text{HR}}$, while contrastively training $\mathbf{I}_{\text{SR}}$ to align with the distribution of $\mathbf{I}_{\text{HR}}$.

To do this, we first prepare two sets of text for image quality, one for low-quality (*e.g.*, $\{bad\}$) and the other for high-quality (*e.g.*, $\{good\}$). Then, we encode them into low-quality text features $\mathbf{Q}_-$ and high-quality text features $\mathbf{Q}_+$, respectively, each in $\mathbb{R}^{1 \times c}$, using the text encoder $\mathcal{E}_{\text{T}}$. Using the quality text features and the image features, we train the encoders to identify $\mathbf{F}_{\text{HR}}$ as higher quality than $\mathbf{F}_{\text{LR}}$ by optimizing the loss function $\mathcal{L}_{\text{LQ}}^{\mathcal{E}}$ as:

$$\mathcal{L}_{\text{LQ}}^{\mathcal{E}} = \text{Cont}(\mathbf{F}_{\text{LR}} \parallel \mathbf{F}_{\text{HR}}, \mathbf{Q}_- \parallel \mathbf{Q}_+). \tag{3}$$

Note that Equation 3 shows the case when the batch size is 1. In our actual training, we extend $\mathbf{Q}_-$ and $\mathbf{Q}_+$ to $N$ batches.

Together with $\mathcal{L}_{\text{LQ}}^{\mathcal{E}}$, we compute the loss function $\mathcal{L}_{\text{LQ}}^{\text{SR}}$ to train the SR network, encouraging the image encoder $\mathcal{E}_{\text{T}}$ to classify $\mathbf{I}_{\text{SR}}$ as $\mathbf{I}_{\text{HR}}$. Unlike Equation 2, where each image feature is matched to a unique content text feature, $\mathcal{L}_{\text{LQ}}^{\text{SR}}$ should ensure that every $\mathbf{I}_{\text{SR}}$ aligns with a single high-quality text feature $\mathbf{Q}_+$. To this end, we modify Equation 2 for $\mathcal{L}_{\text{LQ}}^{\text{SR}}$ as:

$$\mathcal{L}_{\text{LQ}}^{\text{SR}} = -\frac{1}{*} \left( \log \frac{e^{\cos(\mathbf{F}_{\text{SR}}, \mathbf{Q}_+)}}{e^{\cos(\mathbf{F}_{\text{SR}}, \mathbf{Q}_-)}} \right). \tag{4}$$

Similar to Equation 3, note that during actual training, $\mathbf{Q}_-$ and $\mathbf{Q}_+$ are extended to $N$ batches, and $*$ in Equation 4 is set to $N$. Finally, the linguistic-quality loss $\mathcal{L}_{\text{LQ}}$ is defined as the sum of Equations 3 and 4 ($\mathcal{L}_{\text{LQ}} = \mathcal{L}_{\text{LQ}}^{\mathcal{E}} + \mathcal{L}_{\text{LQ}}^{\text{SR}}$).

### 3.5 MODIFIED VGG LOSS

While the proposed linguistic losses, $\mathcal{L}_{\text{LC}}$ and $\mathcal{L}_{\text{LQ}}$, help guide the SR network to produce visually pleasant SR images, relying solely on them may not ensure the preservation of low-frequency information (*e.g.*, structure and color). To address this, we modify the conventional perceptual loss Ledig et al. (2017) $\mathcal{L}_{\text{VGG}}$ and apply it between $\mathbf{I}_{\text{SR}}$ and $\mathbf{I}_{\text{LR}}$ in our LA-SR framework. Given the difference in resolution between $\mathbf{I}_{\text{SR}}$ and $\mathbf{I}_{\text{LR}}$, we incorporate additional pooling steps to extract VGG features for $\mathbf{I}_{\text{SR}}$.

Finally, we combine all the aforementioned loss functions to formulate the total loss $\mathcal{L}_{\text{tot}}$ as follows:

$$\mathcal{L}_{\text{tot}} = \lambda_{\text{LC}} \mathcal{L}_{\text{LC}} + \lambda_{\text{LQ}} \mathcal{L}_{\text{LQ}} + \lambda_{\text{VGG}} \mathcal{L}_{\text{VGG}}, \tag{5}$$

where $\lambda_{\text{LC}} = 0.5$, $\lambda_{\text{LQ}} = 0.5$, and $\lambda_{\text{VGG}} = 0.1$ are empirically chosen hyperparameters of the corresponding loss functions.

Table 1: ×**4 SR performance comparison with un-supervised and self-supervised SR methods on various benchmark datasets.** MASA-SR Lu et al. (2021) and DCSR Wang et al. (2021a) use additional reference images.

| Method | Dataset | BRISQUE↓ | CLIPIQA↑ | TOPIQ↑ | MUSIQ↑ |
|---|---|---|---|---|---|
| DASR Wei et al. (2021) | DRealSR | 45.67 | 0.29 | 0.26 | 27.73 |
| PDD Zhang et al. (2024) | | **27.43** | 0.44 | 0.31 | 37.08 |
| SRTTA Deng et al. (2023) | | 37.06 | 0.42 | 0.32 | 37.22 |
| **LA-SR (Ours)** | | 30.07 | **0.46** | **0.40** | **40.70** |
| MASA-SR Lu et al. (2021) | Camera Fusion | 28.23 | 0.60 | 0.56 | 58.88 |
| DCSR Wang et al. (2021a) | | 34.46 | 0.48 | 0.41 | 55.29 |
| **LA-SR (Ours)** | | **12.47** | **0.69** | **0.58** | **58.97** |
| MASA-SR Lu et al. (2021) | CUFED5 | 9.67 | 0.65 | 0.59 | **67.92** |
| DCSR Wang et al. (2021a) | | 12.88 | 0.59 | 0.54 | 67.71 |
| **LA-SR (Ours)** | | **5.70** | **0.66** | **0.61** | 67.26 |

(a) Bicubic  (b) MASA-SR

(c) DCSR  (d) **LA-SR (Ours)**

Figure 3: **Visual comparison of** ×4 **SR with previous self-supervised SR methods.**

## 4 EXPERIMENTS

**Datasets.** As LA-SR only requires high-quality images, we train our model using high-quality images from DF2K Timofte et al. (2017) and LSDIR Li et al. (2023). For evaluation, we apply SR networks to various benchmarks, including Set5 Bevilacqua et al. (2012), Set14 Zeyde et al. (2010), BSD100 Martin et al. (2001), General100 Dong et al. (2016), Urban100 Huang et al. (2015), and DIV2K Timofte et al. (2017). Unlike previous studies Zhang et al. (2018c); Wang et al. (2021b); Zhang et al. (2021), which focus on bicubic-degraded images, we show results on their natural high-quality images. Moreover, we compare performance on real low-quality images from OST Wang et al. (2018a), DRealSR Wei et al. (2020), and CameraFusion Wang et al. (2021a).

**Metrics.** As our goal is to generate photo-realistic SR images, we focus on evaluating visual quality using non-reference-based perceptual metrics. We employ BRISQUE Mittal et al. (2012), CLIP-IQA Wang et al. (2023a), TOPIQ (NR) Chen et al. (2024), and MUSIQ Liang et al. (2021), which are metrics for assessing perceptual quality in a reference-free setting. Furthermore, while the aforementioned perceptual metrics are central to our evaluation, for completeness, we also report reference-based distortion metrics, such as PSNR and SSIM, even though they are not our main focus. Additionally, we include reference-based perceptual metrics, including LPIPS Zhang et al. (2018a), DISTS Ding et al. (2020), and TOPIQ (FR) Chen et al. (2024).

**Experimental configurations.** Like prior SR methods Zhang et al. (2021); Wang et al. (2021b), we initialize our SR network using PSNR-oriented pre-trained SR models, which are trained to reconstruct HR images from bicubic-downsampled LR images. We then fine-tune the SR network using the LA-SR framework under Equation 5. For training, we use $\mathbf{I}_{LR}$ sized $56 \times 56$ and $\mathbf{I}_{HR}$ sized $224 \times 224$ with a batch size of 16. The model is trained for 300,000 iterations with a fixed learning rate of $5 \times 10^{-5}$, and the entire training process takes 50 hours using four Quadro RTX 8000 GPUs.

### 4.1 COMPARISON WITH PREVIOUS SR METHODS

**Comparison with un- and self-supervised SR methods.** Since LA-SR framework is designed to trained with unpaired low- and high-resolution images, it naturally shares conceptual similarities with both unsupervised and self-supervised SR approaches. To highlight these connections, we first present a comprehensive comparison with prior unsupervised SR methods Wei et al. (2021); Zhang et al. (2024) and self-supervised SR methods Wang et al. (2021a); Lu et al. (2021); Deng et al. (2023) in Table 1. When compared to unsupervised methods Wei et al. (2021); Zhang et al. (2024); Deng et al. (2023), which learn SR without any paired HR supervision, LA-SR consistently delivers higher-quality reconstructions on real-world LR images, demonstrating stronger generalization to natural degradations. Furthermore, against self-supervised methods Wang et al. (2021a); Lu et al. (2021), which typically rely on reference images or synthetic degradations to guide the learning process, LA-SR achieves competitive or superior performance while requiring no additional reference inputs. This highlights LA-SR's ability to leverage language-based supervision to bridge the gap between unpaired LR-HR domains, achieving effective real-world SR without the constraints of explicit pairing or handcrafted degradations. Moreover, Figure 3 provides visual comparisons. As shown, previous self-supervised approaches Lu et al. (2021); Wang et al. (2021a) struggle to accurately recover fine text details, whereas LA-SR produces sharper and more readable results.

Table 2: ×4 **SR performance comparison with supervised SR methods on various benchmark datasets.** ESRGAN and ESRGAN† are trained on bicubic Timofte et al. (2017) and real Wei et al. (2020) LR images, respectively.

| Method | Dataset | BRISQUE↓ | CLIPIQA↑ | TOPIQ↑ | MUSIQ↑ | Dataset | BRISQUE↓ | CLIPIQA↑ | TOPIQ↑ | MUSIQ↑ |
|---|---|---|---|---|---|---|---|---|---|---|
| ESRGAN Wang et al. (2018b) | Set5 | 61.08 | 0.53 | 0.31 | 53.35 | Set14 | 54.21 | 0.51 | 0.37 | 46.17 |
| ESRGAN† | | 32.13 | 0.47 | 0.33 | 58.97 | | 40.08 | 0.46 | 0.35 | 51.43 |
| RealESRGAN Wang et al. (2021b) | | 24.20 | 0.52 | 0.40 | 64.65 | | 28.30 | 0.52 | 0.44 | 57.02 |
| **LA-SR (Ours)** | | **11.53** | **0.65** | **0.52** | **67.07** | | **10.13** | **0.64** | **0.53** | **60.04** |
| ESRGAN | BSD100 | 53.45 | 0.54 | 0.34 | 48.47 | General100 | 54.66 | 0.55 | 0.33 | 48.09 |
| ESRGAN† | | 41.35 | 0.43 | 0.35 | 51.18 | | 36.15 | 0.50 | 0.37 | 55.71 |
| RealESRGAN | | 24.32 | 0.50 | 0.44 | 57.83 | | 32.22 | 0.54 | 0.44 | 59.20 |
| **LA-SR (Ours)** | | **5.71** | **0.68** | **0.58** | **64.50** | | **19.31** | **0.63** | **0.52** | **62.76** |
| ESRGAN | Urban100 | 46.75 | 0.54 | 0.41 | 30.39 | DIV2K | 54.16 | 0.53 | 0.32 | 34.86 |
| ESRGAN† | | 46.83 | 0.45 | 0.37 | **33.84** | | 40.86 | 0.42 | 0.35 | 40.22 |
| RealESRGAN | | 21.51 | 0.59 | 0.55 | 32.85 | | 25.72 | 0.50 | 0.44 | 45.84 |
| **LA-SR (Ours)** | | **7.20** | **0.71** | **0.61** | 30.93 | | **10.90** | **0.62** | **0.53** | **46.31** |
| ESRGAN | OST | 57.40 | 0.50 | 0.45 | 35.25 | DRealSR | 69.18 | 0.42 | 0.22 | 22.36 |
| ESRGAN† | | 48.78 | 0.39 | 0.35 | 43.13 | | 28.07 | 0.49 | 0.35 | 35.25 |
| RealESRGAN | | 11.68 | 0.54 | 0.56 | **62.06** | | 30.31 | **0.51** | **0.46** | 40.02 |
| **LA-SR (Ours)** | | **7.07** | **0.64** | **0.59** | 60.82 | | **30.07** | 0.46 | 0.40 | **40.70** |
| ESRGAN | Dataset Diffusion | 66.10 | 0.56 | 0.27 | 4.52 | LDM | 55.96 | 0.61 | 0.30 | 4.35 |
| ESRGAN† | | 33.22 | 0.48 | 0.35 | 4.65 | | 34.32 | 0.51 | 0.34 | 4.57 |
| RealESRGAN | | 19.06 | 0.60 | 0.50 | 4.91 | | 25.22 | 0.57 | 0.43 | 4.66 |
| **LA-SR (Ours)** | | **4.68** | **0.65** | **0.51** | **4.95** | | **6.22** | **0.67** | **0.50** | **4.80** |

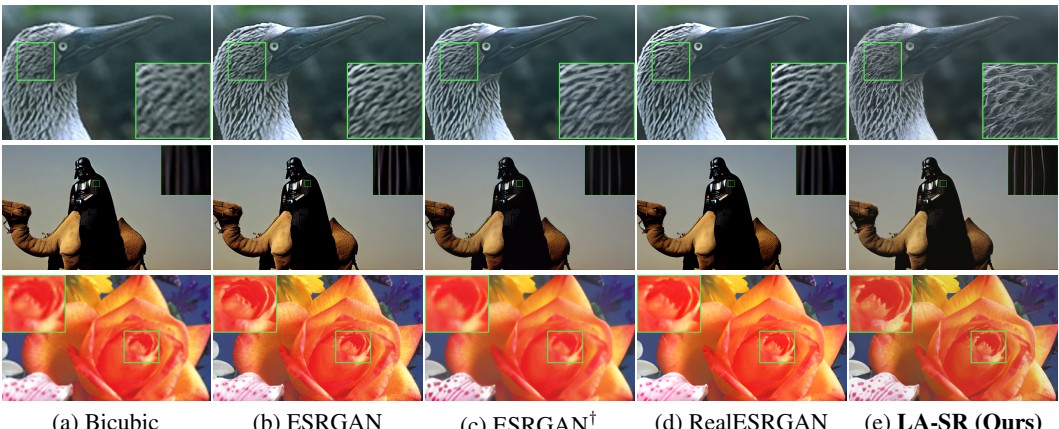

| (a) Bicubic | (b) ESRGAN | (c) ESRGAN† | (d) RealESRGAN | (e) **LA-SR (Ours)** |
|---|---|---|---|---|

Figure 4: **Visual comparison of ×4 SR on benchmark datasets with previous supervised SR methods.** (Zoom-in for best view)

**Comparison with supervised SR methods.** Furthermore, we extend our comparison to supervised SR methods that are trained with explicit LR–HR pairs constructed in different ways. In particular, using the same RRDB backbone Wang et al. (2018b) as our SR network for a fair evaluation, we compare LA-SR against three representative settings: (1) methods trained on bicubic-downsampled LR images Wang et al. (2018b), (2) methods trained on real LR images captured using specialized camera (denoted as ESRGAN†), and (3) methods trained on synthetically degraded LR images with more complex degradation models Wang et al. (2021b) setups Wei et al. (2020).

First, as shown in Table 2, ESRGAN Wang et al. (2018b) trained on bicubic-downsampled degraded images consistently performs the worst due to a large gap between its training and testing domains. Similarly, ESRGAN†, despite being trained on real-degraded images Wei et al. (2020) from specific camera models, shows limited generalizability on other images. Likewise, models Wang et al. (2021b) trained on synthetic datasets often struggle to super-resolve high-quality images. Second, we evaluate performance on real low-quality images using the OST Wang et al. (2018a) and DRealSR Wei et al. (2020) datasets. Since LA-SR is trained on naturally degraded LR images influenced by subject-to-camera distance, it also performs well on real low-quality images.

Moreover, Figure 4 provides visual comparisons across different supervised methods. As shown, even when compared with supervised methods, LA-SR consistently reconstructs finer structures, such as hair strands and subtle textures (*i.e.*, the texture of hair and flower in the first and last row of Figure 4), yielding more natural and detailed outputs. For further analysis of another key aspect (*i.e.*, distortion reduction), please refer to Appendix C. For more visual comparisons, please refer to Appendix G.

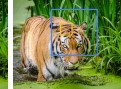 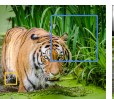 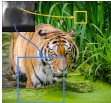

(a) Overlap    (b) Inclusion    (c) Random    (d) **Ours**

Figure 5: **Different approaches to preparing $\mathbf{I}_{LR}$ and $\mathbf{I}_{HR}$.** In our approach, $\mathbf{I}_{LR}$ and $\mathbf{I}_{HR}$ are selected based on the depth.

Table 3: **Comparison of models on DIV2K validation set under various LR-HR preparation approaches.**

| Method | BRISQUE↓ | CLIPIQA↑ | TOPIQ↑ | MUSIQ↑ |
|---|---|---|---|---|
| Overlap | 13.72 | 0.59 | 0.51 | 42.10 |
| Inclusion | 11.76 | 0.59 | 0.50 | 42.09 |
| Random | **8.45** | 0.61 | 0.51 | 42.16 |
| **Depth-based (Ours)** | 10.90 | **0.62** | **0.53** | **46.31** |

Table 4: **Quantitative comparison with background filtering.**

Table 5: **Quantitative comparison with adding synthetic degradations.**

| Background filtering | BRISQUE↓ | CLIPIQA↑ | TOPIQ↑ | MUSIQ↑ |
|---|---|---|---|---|
| ✓ | **7.15** | 0.61 | 0.52 | **46.66** |
| ✗ (Ours) | 10.90 | **0.62** | **0.53** | 46.31 |

| | Real38 | | | DRealSR | | |
|---|---|---|---|---|---|---|
| LR image sourced | CLIPIQA↑ | TOPIQ↑ | MUSIQ↑ | CLIPIQA↑ | TOPIQ↑ | MUSIQ↑ |
| Depth + RealESRGAN | **0.66** | 0.54 | **66.76** | 0.42 | 0.38 | 39.12 |
| **Depth (Ours)** | 0.63 | **0.56** | 63.48 | **0.46** | **0.40** | **40.70** |

**Comparison on generated images.** With the increasing prevalence of AI-generated images, the ability to perform SR on such synthetic content has become increasingly important. To evaluate this capability, the lower portion of Table 2 shows SR performance on synthetically generated images, including the Dataset Diffusion Nguyen et al. (2024) and a curated set of 100 images generated by LDM Rombach et al. (2022). As shown, LA-SR consistently delivers superior performance not only on real-world degraded images but also on these high-quality synthetic samples. This demonstrates that LA-SR effectively generalizes across both natural and diffusion-generated domains.

## 4.2 EFFECT OF DEPTH-BASED LR-HR PREPARATION

**Comparison with differently prepared LR-HR patches.** Table 3 compares different approaches for obtaining LR-HR patches. While we use depth to assign distant patches as $\mathbf{I}_{LR}$ and closer patches as $\mathbf{I}_{HR}$, we also evaluate three alternatives: overlap, inclusion, and random. In the overlap (Figure 5a), $\mathbf{I}_{HR}$ is randomly chosen and $\mathbf{I}_{LR}$ is selected to overlap with it beyond a certain threshold (*i.e.*, 0.4). On the other hand, in the inclusion (Figure 5b), $\mathbf{I}_{HR}$ is randomly selected and $\mathbf{I}_{LR}$ is confined within HR patches. As shown in the table, both the overlap and inclusion approaches result in excessive similarity between $\mathbf{I}_{LR}$ and $\mathbf{I}_{HR}$, making differentiation challenging. In the random (in Figure 5c), $\mathbf{I}_{LR}$ and $\mathbf{I}_{HR}$ are selected randomly. While the random approach performs better than overlap and inclusion, our depth-based approach (Figure 5d) consistently delivers more reliable results.

**Background filtering.** To train LA-SR, we extract LR from distant depth. However, since background areas are typically located farther from the camera, this depth-based sampling strategy tends to select patches that are dominated by background content, which may contain fewer meaningful structures or fine details compared to foreground regions. To examine the impact of this potential bias, we conduct an empirical study comparing a naive sampling strategy with a background filtering approach in Table 4. As shown, while filtering out background regions provides a slight performance gain, background patches still contribute valuable contextual cues that benefit super-resolution training.

**Distance-Irrelevant Degradations.** Since the real LR patches $\mathbf{I}_{LR}$ used in our LA-SR framework are primarily derived from naturally captured images, they may not fully represent extreme degradations such as those found in animation frames or old film footage. To improve robustness under sevely degraded conditions, we incorporate synthetic degradations following the strategy of RealESRGAN. As shown in Table 5, introducing such synthetic degradations significantly enhances performance on heavily degraded datasets, especially Real38 dataset, which includes diverse and challenging artifacts. However, this augmentation also introduces a trade-off: it causes a slight performance drop on datasets like DRealSR, where the degradations are primarily natural (*e.g.*, distance-based blur) and lack the complex, domain-specific artifacts modeled by synthetic processes.

## 4.3 EFFECT OF THE PROPOSED LOSS FUNCTIONS

Table 6 shows a quantitative analysis of our proposed loss functions: $\mathcal{L}_{LC}$, $\mathcal{L}_{LQ}$, and $\mathcal{L}_{VGG}$, by evaluating performance when each term is ommitted. While using all loss terms consistently yields the best perceptual performance, Figure 6 further shows the individual contributions of each loss function. First, omitting $\mathcal{L}_{LQ}$ (Figure 6a) results in severe artifacts. This likely occurs because

Table 6: **Comparison of models on DIV2K under various loss combinations.** The markers ✓ and ✗ indicate whether the corresponding loss is applied or not, respectively.

| $\mathcal{L}_{LC}$ | $\mathcal{L}_{LQ}$ | $\mathcal{L}_{VGG}$ | BRISQUE↓ | CLIPIQA↑ | TOPIQ↑ | MUSIQ↑ |
|---|---|---|---|---|---|---|
| ✓ | ✗ | ✓ | 38.42 | 0.35 | 0.31 | 30.99 |
| ✗ | ✓ | ✓ | 14.94 | 0.58 | 0.50 | 42.70 |
| ✓ | ✓ | ✗ | **10.32** | 0.59 | 0.50 | 37.41 |
| ✓ | ✓ | ✓ | 10.90 | **0.62** | **0.53** | **46.31** |

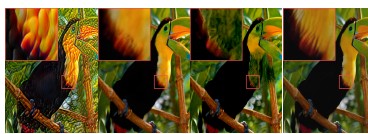

(a) w/o $\mathcal{L}_{LQ}$  (b) w/o $\mathcal{L}_{LC}$  (c) w/o $\mathcal{L}_{VGG}$  (d) **Ours**

Figure 6: **Visual comparison under different loss combinations.** The underlying caption refers to the omitted loss terms during training.

multiple images can match the same content description (*e.g.*, different images could correspond to "a bird" while still satisfying perceptual loss). Therefore, without $\mathcal{L}_{LQ}$, the network lacks a strong regularizer to discourage implausible high-frequency details, leading to noisy or distorted reconstructions. Similarly, omitting $\mathcal{L}_{LC}$ (Figure 6b) causes the network to generate irrelevant details. Without explicit content alignment, the reconstruction drifts away from the intended structure, producing objects that are irrelevant to the input. Moreover, omitting $\mathcal{L}_{VGG}$ (Figure 6c) hinders the network's ability to recover low-frequency details, causing noticeable color shifts. In contrast, our full model (Figure 6d) trained with all three loss components, successfully balances structural fidelity, perceptual realism, and semantic alignment.

### 4.4 VISUALIZATION OF ENCODED FEATURES

Figure 7 provides a visual analysis of the encoded features to show the role of our encoders. Figure 7a shows a t-SNE Van der Maaten & Hinton (2008) projection of two image features $\mathbf{F}_{LR}$ and $\mathbf{F}_{HR}$. As shown, our encoders effectively distinguish between $\mathbf{F}_{LR}$ and $\mathbf{F}_{HR}$, mapping them into the separate spaces (*e.g.*, **red** and **blue** clusters). This indicates that encoders classify LR and HR images based on quality, enabling meaningful backpropagation in $\mathcal{L}_{LQ}^{SR}$. Figure 7b shows the distribution of features $\mathbf{F}_{HR}$, where images with the same color share the same keywords in their context texts. For example, the image-to-text model Li et al. (2022) produces context texts that include 'Water' for all the **purple** images. Interestingly, while the majority of images are distinctly divided by keywords, indicating that our encoders organize images according to their corresponding content, the **orange** and **yellow** patches are positioned closer together, reflecting similar properties between 'Cat' and 'Dog'.

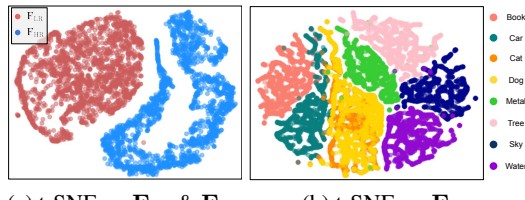

(a) t-SNE on $\mathbf{F}_{LR}$ & $\mathbf{F}_{HR}$ (b) t-SNE on $\mathbf{F}_{HR}$

Figure 7: **Distribution of encoded image features.** Each point represents a t-SNE projection of features: (a) Our encoders effectively distinguish between LR and HR patches. (b) Within the HR patches, features are well-clustered into distinct spaces, capturing contextual information.

## 5 LIMITATIONS

While LA-SR shows strong SR performance (Tables 1, 2, and Figure 4), our LR patches are mainly extracted from high-quality images using depth information. For cases like animation or old films, different unseen degradation patterns can cause ringing and artifacts (also described in Section 4.2). We plan to expand the dataset to cover more diverse degradations in future work.

## 6 CONCLUSION

In this work, we present LA-SR, a novel SR framework that uses real LR images naturally occurring in photographs due to subject-camera distance. To handle the challenge of unpaired LR-HR images in our framework, we present linguistic-content and linguistic-quality losses, which guide content preservation and quality—two key factors in SR. Extensive experiments show that LA-SR achieves superior SR performance across various images. Also, our framework can be extended to other image restoration tasks, including deblurring and denoising.

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

## A  APPENDIX

In this appendix, we provide discussions, details, and more visual results that could not be included in the main manuscript due to lack of space.

## B  COMPARISON WITH DIFFUSION-BASED SR

In the main paper, we ensure a fair comparison by evaluating our LA-SR against methods that employ the same RRDB architecture. Table 7 further compares LA-SR with recent diffusion-based super-resolution approaches, including LDM Rombach et al. (2022), StableSR Wang et al. (2024), and SeeSR Wu et al. (2024). As shown, although diffusion-based methods generally achieve stronger perceptual quality, our LA-SR attains competitive results, performing on par with LDM and StableSR and even surpassing them in several non-reference metrics (*e.g.*, TOPIQ (NR) and MUSIQ). Considering the substantially lower parameter count and faster runtime of LA-SR compared to diffusion-based approaches, these results highlight its favorable balance between efficiency and performance.

Table 7: ×4 SR performance comparison with diffusion-based SR methods on DRealSR.

| Method | BRISQUE$_\downarrow$ | CLIPIQA$_\uparrow$ | TOPIQ (NR)$_\uparrow$ | MUSIQ$_\uparrow$ | Params (M)$_\downarrow$ | Runtime$_\downarrow$ |
|---|---|---|---|---|---|---|
| LDM | 35.48 | 0.47 | 0.32 | 39.26 | 113.62 | 18.9 |
| StableSR | 44.43 | 0.43 | 0.32 | 40.31 | 149.91 | 27.47 |
| SeeSR | 27.76 | 0.51 | 0.44 | 41.60 | 2524 | 38.70 |
| **LA-SR (Ours)** | 30.07 | 0.46 | 0.40 | 40.70 | 16.7 | 0.6 |

## C  DISTORTION-BASED COMPARISON

Table 8 presents various reference-based metrics, including distortion-based measures (*e.g.*, PSNR and SSIM) and perceptual metrics (*e.g.*, LPIPS Zhang et al. (2018a), DISTS Ding et al. (2020), and TOPIQ (FR) Chen et al. (2024)). Although LA-SR may not achieve the highest distortion-based metric scores due to the trade-off between perceptual and distortion quality under limited network capacity Blau & Michaeli (2018), it remains competitive and excels in DISTS and TOPIQ (FR).

Table 8: **Reference-based** ×4 **SR performance comparison on real LR Wei et al. (2020) dataset.**

| Method | PSNR$_\uparrow$ | SSIM$_\uparrow$ | LPIPS$_\downarrow$ | DISTS$_\downarrow$ | TOPIQ (FR)$_\uparrow$ |
|---|---|---|---|---|---|
| RealESRGAN Wang et al. (2021b) | **25.84** | **0.80** | **0.28** | **0.15** | 0.29 |
| StableSR Wang et al. (2024) | 24.22 | 0.70 | 0.36 | 0.18 | 0.24 |
| **LA-SR (Ours)** | 25.59 | 0.78 | 0.30 | **0.15** | **0.30** |

## D  RELIANCE ON DEPTH ESTIMATOR

We construct the unpaired LR-HR pairs using depth information. Since the accuracy of the predicted depth depends on the depth estimator used, we provide a quantitative comparison across different estimators. As shown in Table 9, our LA-SR demonstrates robust performance regardless of the chosen depth estimator, indicating that substituting DepthFM with other networks, Depth Anything V2 Yang et al. (2024), yields comparable results on DIV2K.

Table 9: **Quantitative comparison with using different depthmap estimator.**

| Method | BRISQUE$_\downarrow$ | LIQE$_\uparrow$ | CLIPIQA$_\uparrow$ | TOPIQ$_\uparrow$ | MUSIQ$_\uparrow$ |
|---|---|---|---|---|---|
| Depth Anything V2 | 11.81 | 3.17 | **0.63** | **0.53** | **46.44** |
| DepthFM | **10.90** | **3.18** | 0.62 | **0.53** | 46.31 |

# E PRACTICAL APPLICATIONS

## E.1 FINE-TUNING ON SPECIFIC IMAGES

As LA-SR framework operates in an unpaired setting, it allows training directly on the target image intended for super-resolution. To demonstrate this, we randomly sample 10 images and fine-tune our SR network on these images, with evaluations shown on the same set in Figure 8. As shown, the quantitative progression over iterations shows consistent perceptual improvements with fine-tuning. This occurs because our SR network can adapt to specific degradation patterns present in the given images, further enhancing the results. Due to the small size of the dataset, we use a learning rate of $1 \times 10^{-7}$, training only the SR network while keeping the encoders fixed.

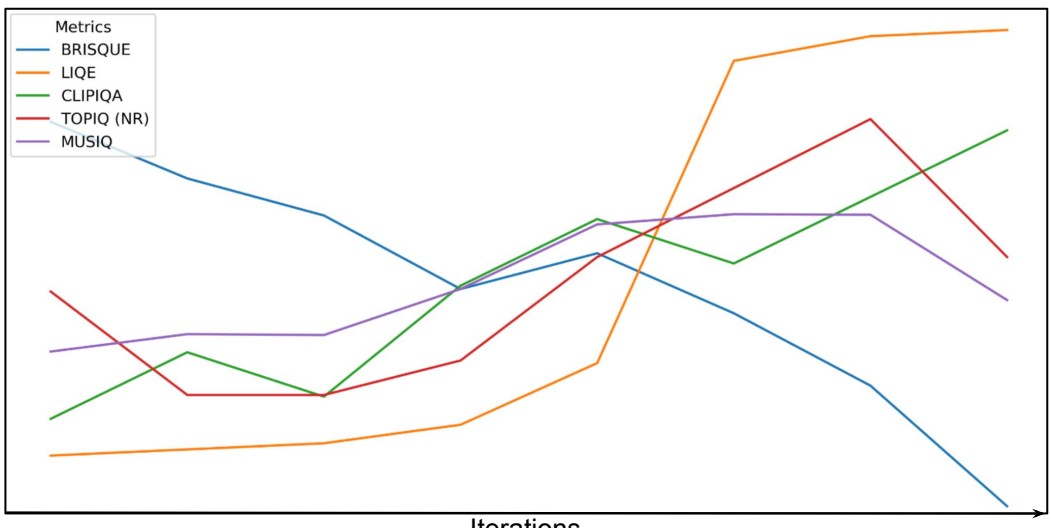

Figure 8: **Quantitative comparison across fine-tuning.** We select 10 images from open databases and fine-tune our SR network on them, assessing metrics on the same images.

## E.2 ITERATIVE APPROACH.

Unlike other methods that train on LR patches that are *always* degraded, LA-SR includes a range of LR patches, from severely degraded images to relatively high-quality ones when all subjects in the image are closer to the camera. As a result, while previous methods are specifically tailored to degraded images, LA-SR remains effective even on high-quality images. Based on this, considering conventional SR methods mapping low-quality images to high-quality ones, LA-SR can be iteratively applied to its own super-resolved outputs, further enhancing image quality.

To demonstrate this, we apply $\times 4$ SR networks twice to generate $\times 16$ SR results in Figure 9. As shown, previous methods Wang et al. (2018b; 2024) face challenges due to the significant domain gap between their training LR images and the recovered SR images, which often produces blurry results. Furthermore, the diffusion-based StableSR Wang et al. (2024) alters the original image structure (*e.g.*, transforming stars into circles in the second row of Figure 9). In contrast, LA-SR shows the most visually appealing results.

# F DETAILS OF IMAGE-TO-TEXT MODEL

We utilize an image-to-text model to extract content texts from images. Specifically, we employ the CLIP Radford et al. (2021)-Interrogator, which evaluates the correlation between words in a dictionary and a given image, selecting the top-$k$ ($k = 24$) most highly correlated words. However, since computing correlations between all words and images is computationally expensive, we manually curate a set of 1,000 representative texts in our dictionary to reduce the overall computational burden.

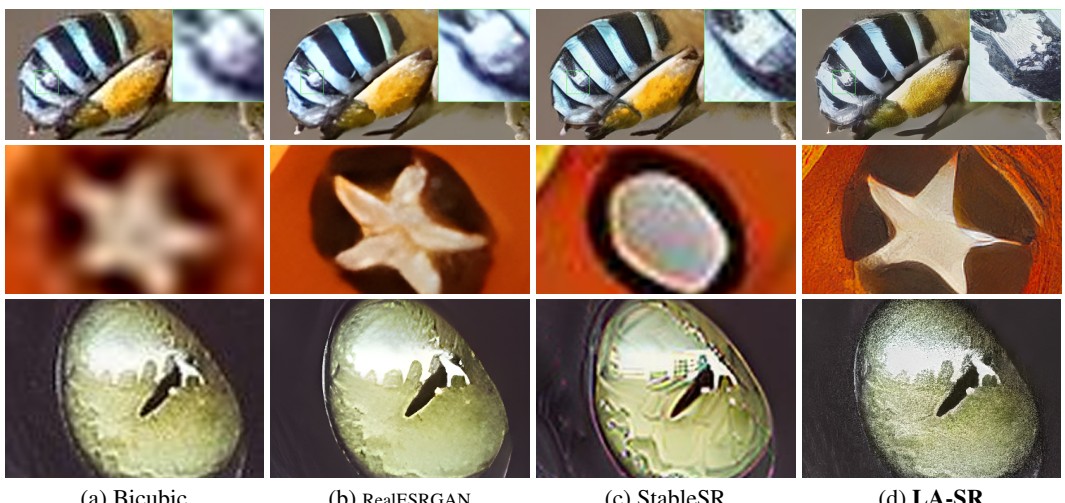

|  |  |  |  |
|---|---|---|---|
| (a) Bicubic | (b) RealESRGAN | (c) StableSR | (d) **LA-SR** |

Figure 9: **Visual comparison of** $\times 16$ **SR with previous SR methods.** To this end, we apply $\times 4$ SR networks twice to images sourced from open databases. (Zoom-in for best view)

## G  ADDITIONAL VISUAL COMPARISON

Figures 10, 11, and 12 present additional visual comparisons between our LA-SR and previous SR methods. The results demonstrate that LA-SR consistently generates perceptually superior outputs compared to existing approaches.

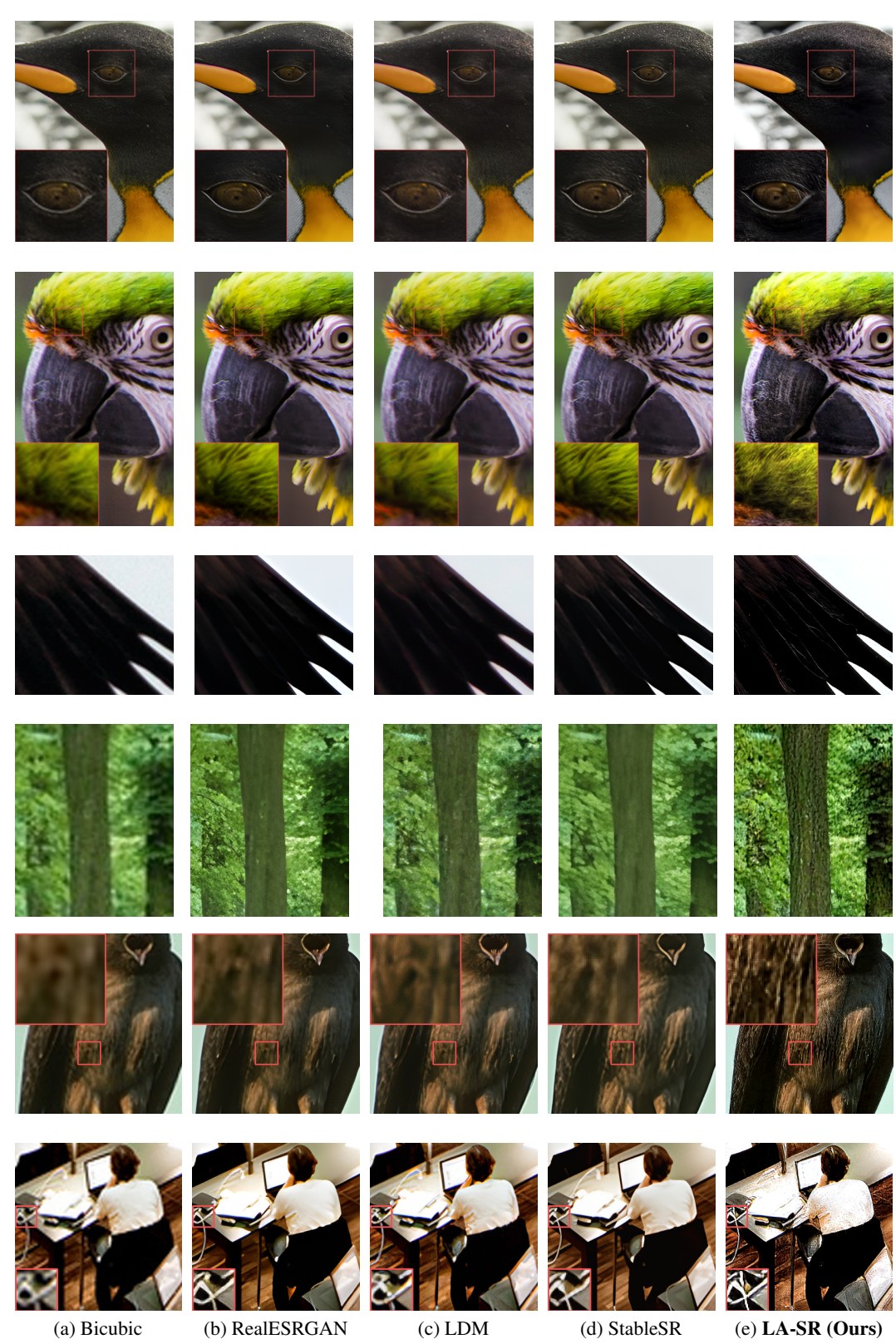

(a) Bicubic     (b) RealESRGAN     (c) LDM     (d) StableSR     (e) **LA-SR (Ours)**

Figure 10: **Visual comparison on** $\times 4$ **SR with previous SR methods.** (Zoom-in for best view)

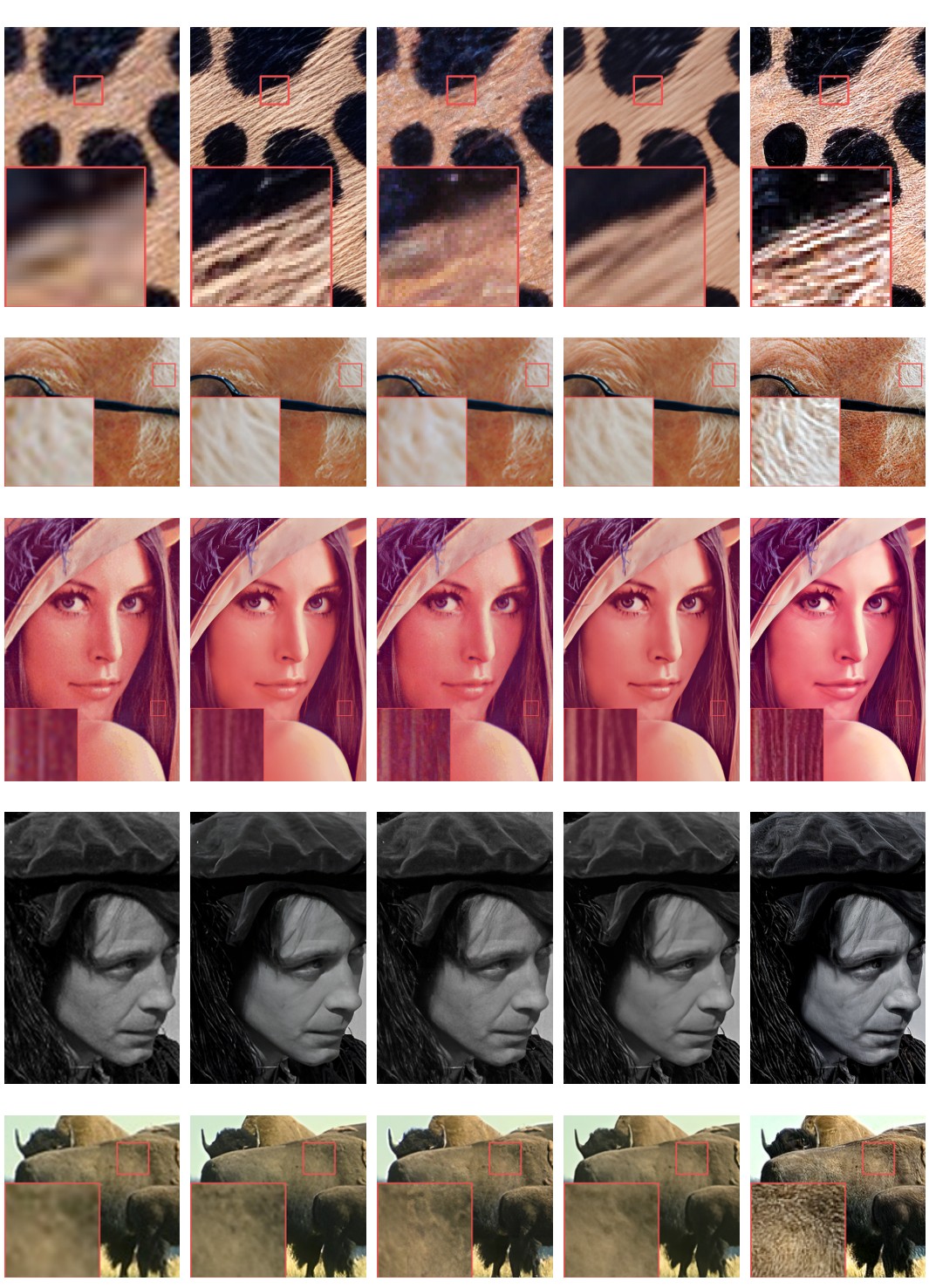

(a) Bicubic     (b) RealESRGAN     (c) LDM     (d) StableSR     (e) **LA-SR (Ours)**

Figure 11: **Visual comparison on** $\times 4$ **SR with previous SR methods.** (Zoom-in for best view)

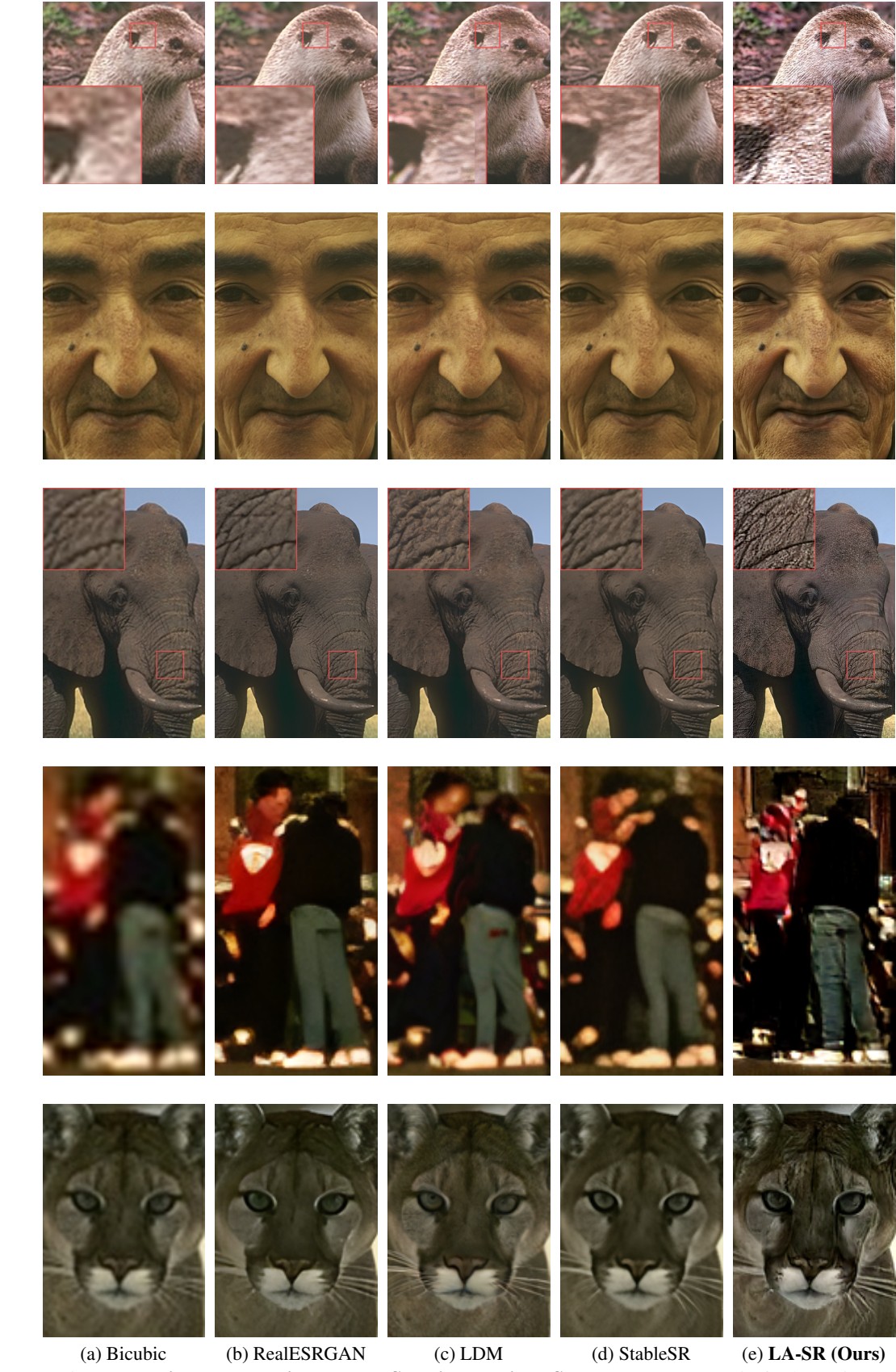

(a) Bicubic     (b) RealESRGAN     (c) LDM     (d) StableSR     (e) **LA-SR (Ours)**

Figure 12: **Visual comparison on** $\times 4$ **SR with previous SR methods.** (Zoom-in for best view)

