# OpenReview forum: "Language-Assisted Super-Resolution from Real-World Low-Resolution Patches"
_ICLR.cc/2026/Conference — Submitted to ICLR 2026_

### Official Review · Reviewer_RjqZ · 2025-10-23

**Soundness:** 2
**Presentation:** 3
**Contribution:** 2
**Rating:** 2
**Confidence:** 3

**Summary:**

This paper tackles the challenge of single image super-resolution (SISR), which reconstructs high-resolution (HR) images from low-resolution (LR) ones, but without relying on paired training data. The authors propose LA-SR (Language Assistant for SR), a novel unpaired super-resolution framework that leverages vision-language models. By observing that different depth regions within a single real image naturally vary in resolution, the approach extracts realistic LR patches from real images. LA-SR maps images into a semantic language space and introduces two language-guided losses: one preserving semantic content and another enhancing perceptual quality. Experimental results demonstrate the effectiveness of the proposed method.

**Strengths:**

Originality: The task of learning from unpaired LR and HR patches extracted from high-quality images sounds original to me.

Significance: If the proposed method really works, it would create a new approach for SISR.

**Weaknesses:**

Clarity: Although the English is good, the proposed method is a bit unclear. Specifically,
- How could the image-to-text model perform reliably across all patches, especially those that only contain textures without an object? And what about those of flat colors (e.g., in man-made objects, fonts, etc.)?
- Even if it can extract text from the patches, how could it help ensure the "content" of the patches is the same? For example, for the text "A seagull sitting on the beach", there are infinite different images with completely different seagulls but all match the same content.
- This also applies to the linguistic-quality loss that different images can all be "good" or "bad".

In general, since languages are very high-level concepts that have very little information about image details, I do not understand how the proposed method could work.

Quality:
- The method is inspired by DGDML-SR (Cheng et al. 2020), why not compare with it?
- Why are most of the comparisons performed on very early baselines (only one in 2024 and the rest are before 2021)? Without comparison to recent methods, it is difficult to evaluate how much improvement the proposed method brings to SISR.
- Why is the proposed method only evaluated on the early RRDB backbone network? The proposed method is agnostic to the network architecture, so why not show its generalizability to state-of-the-art methods?
- The results show that the proposed method seems to be biased towards images with more details (which may be the reason why it performs well on non-reference metrics), but there is no evaluation on whether these details are correct.
- The input LR images are not included.

**Questions:**

Please see my comments above.

---

### Official Review · Reviewer_r1tK · 2025-10-31

**Soundness:** 3
**Presentation:** 3
**Contribution:** 2
**Rating:** 4
**Confidence:** 3

**Summary:**

This paper introduces LA-SR, a language-assisted super-resolution framework designed to handle real-world image degradations without relying on paired LR-HR datasets. The key insight is to extract LR and HR patches from a single high-quality image using a depth estimator—distant regions as LR and closer regions as HR. The method leverages a pretrained vision-language model (CLIP) to align image and language representations via two novel losses: a linguistic-content loss to preserve semantics and a linguistic-quality loss to enhance perceptual realism. Extensive experiments on multiple benchmarks demonstrate that LA-SR achieves competitive or superior performance compared to recent self-supervised, unsupervised, and even some supervised SR methods. Ablation studies validate the design choices, including patch selection strategies and loss components.

**Strengths:**

1. The paper addresses a critical and underexplored challenge in SR: handling real-world degradations without synthetic data or paired supervision. The use of depth-based patch extraction from natural images is both practical and innovative.
2. The integration of vision-language models (CLIP) to bridge the LR-HR gap in an unpaired setting is timely and well-motivated, leveraging recent advances in multimodal learning.
3. The paper provides extensive experiments across diverse benchmarks (e.g., Set5, Urban100, DRealSR, OST) and compares against a wide range of methods (unsupervised, self-supervised, supervised). The consistent superiority in no-reference perceptual metrics (CLIPIQA, TOPIQ, MUSIQ) is compelling.

**Weaknesses:**

1. The reliance on monocular depth estimation for patch extraction is not thoroughly justified. While Table 3 compares alternative strategies, the assumption that depth directly correlates with degradation is oversimplified and ignores other factors (e.g., lens aberrations, motion blur, sensor noise). The robustness to depth estimation errors is only briefly addressed in Appendix D.
2. The paper focuses primarily on GAN-based SR methods and omits comparisons with recent diffusion-based approaches (e.g., SinSR, StableSR) and other blind/unpaired SR methods (e.g., unfolding-based or internal learning methods). This limits the contextualization of LA-SR’s novelty and performance.
3. The paper's depth-based strategy for extracting LR and HR patches is geometrically motivated but semantically agnostic. This raises a significant concern: there is no guarantee that the selected patches contain semantically complete objects (e.g., a full animal or a whole building). A patch from a distant region might only capture a fragment of an object.
4. While LA-SR excels in perceptual metrics, its performance on distortion-based metrics (PSNR, SSIM) is weaker (Table 8). The trade-off is acknowledged but not deeply analyzed, raising concerns about potential overfitting to CLIP-based metrics.

**Questions:**

1. Integrating LA-SR’s language-guided losses into diffusion-based SR frameworks? Could such a combination further enhance performance or robustness?
2.  A more detailed analysis of LA-SR’s sensitivity to depth estimation errors? Are there common failure cases (e.g., inaccurate depth in textured or reflective surfaces) and how do they impact SR performance?

---

### Official Review · Reviewer_PnCK · 2025-10-31

**Soundness:** 2
**Presentation:** 3
**Contribution:** 2
**Rating:** 2
**Confidence:** 5

**Summary:**

This paper introduces LA-SR (Language Assistant for SR), a novel framework for unpaired real-world image super-resolution. The work is motivated by the difficulty of collecting paired real-world LR-HR data and the significant domain gap of synthetic degradations. The authors' key insight is that a single high-quality image naturally contains real-world LR patches (distant objects) and HR patches (near objects) due to varying depths of field, thus providing a source for unpaired, realistic training data.

**Strengths:**

1.  The paper is exceptionally well-written and structured. Figure 1, in particular, provides a highly intuitive and concise overview of the entire framework, effectively communicating the core idea.
2. The strategy of mining unpaired, real-world LR-HR patches from single images using depth is highly practical and scalable. It cleverly bypasses the need for specialized hardware or complex synthetic degradation pipelines, giving it significant real-world applicability.

**Weaknesses:**

1. The framework's core assumption—that distance directly correlates with image quality ('far = low-res', 'near = high-res')—is an oversimplification of real-world imaging physics. This link can be broken by factors like motion blur on near objects, simple textures in the distance (e.g., a clear sky), or the camera's focal plane. Treating depth as the sole proxy for quality is a major limitation.
2. The reliance on the depth-quality assumption may lead to sampling biases. For instance, the method might struggle to extract effective LR-HR pairs from images with a shallow depth of field (e.g., portraits, flat surfaces). This could result in a training dataset that lacks certain scene types, potentially harming the model's generalization.
3. The use of binary, coarse-grained quality labels like {good} and {bad} is a simplistic form of supervision. This may not capture the full spectrum of real-world degradations (e.g., noise, compression artifacts) and could limit the model's ability to handle complex, unseen degradations beyond simple blurriness.
4. While the paper focuses on perceptual quality, the reference-based metrics reported in the appendix (e.g., PSNR/SSIM) are not consistently strong compared to other methods. This suggests a significant trade-off where the model gains perceptual realism at the cost of reconstruction fidelity.

**Questions:**

1. How does your data curation method handle scenes where depth does not strongly correlate with quality (e.g., motion blur on near objects, or sharp, distant textures)? Does this introduce significant label noise into your training set?
2. How does the patch extraction algorithm perform on images with a shallow depth of field (e.g., portraits, documents)? Is there a risk of failing to extract a sufficient number or diversity of LR/HR patches from such images?
3. Could you provide more details on the "pre-defined quality texts"? Was a richer vocabulary used beyond binary labels like {good}/{bad} to describe different quality dimensions (e.g., 'sharp', 'noisy', 'blurry')?
4. The results in the appendix show that LA-SR does not always perform well on reference-based metrics like PSNR. Could you comment on this trade-off between perceptual quality and reconstruction fidelity within your framework?

---

> ### Comment · Reviewer_PnCK · 2025-11-28
> **Rebuttal**
>
> Currently, I have assigned a score of 2, which is consistent with the assessment of Reviewer RjqZ. I would like to emphasize that this evaluation is the result of a careful and objective review of the paper based on its technical merits. However, I remain open-minded and am willing to raise my score if the authors can effectively address the key concerns I have raised in their rebuttal.

---

### Official Review · Reviewer_89QH · 2025-11-04

**Soundness:** 2
**Presentation:** 3
**Contribution:** 3
**Rating:** 6
**Confidence:** 4

**Summary:**

This work presents a method for performing super-resolution using CLIP. Specifically, it utilizes a cross-modal model to convert image features into text features, which consist of two parts: content and quality. By constraining the consistency of the content and the quality classification loss of high-resolution and low-resolution patches. He forced the network to distinguish between the original low-resolution patches and the high-resolution patches of the same image, ensuring that they were consistent with the high and low quality characteristics of the text. Overall, I think this work is innovative. Super-resolution is achieved by varying the resolution of different patches in the same image, and processing the mismatch between patches through semantics is of great reference value.

**Strengths:**

- This work focuses on the issue of super-resolution for unmatched images. Unlike the traditional end-to-end approach, it actually breaks down this problem into two parts: content consistency loss and quality improvement loss. Since the content is generated by image2text, it can avoid the original requirement for paired images.
- The organization of the paper is excellent, with a clear and understandable structure. I really enjoyed reading it.
- The experiment is complete and the ablation experiments are thorough.

**Weaknesses:**

- The text references the use of CLIP for both the image and text encoders, including modified positional embeddings, but thorough discussion of the possible limitations of frozen CLIP features (e.g., out-of-domain performance or cross-resolution consistency) is lacking. The paper does not quantify how much the backbone choice, or the specifics of the dictionary/prior texts, influences the SR result. This could mask some confounding factors in the comparison to other approaches.
- The selection of the baseline is old.

**Questions:**

- Considering that the consistency of the content is achieved through image2text generation, I doubt the effectiveness of this approach for some patches with complex content. This is my main concern. Compared to purely visual SR methods, the reliability of maintaining content consistency through text is rather questionable.
- How does the choice of language model backbone (e.g., CLIP vs. BLIP, or other multimodal encoders) affect downstream SR performance? Were alternatives or diverse classifier/dictionary setups tested?
- The model requires the updated Backbone for comparison.
- How robust is the model to text prompts that represent high-quality and low-quality images?

---

### Meta-Review · Area_Chair_UyTs · 2026-01-06

**Summary:**

The reviewers generally see this paper as tackling an important and practically meaningful problem, namely, real-world image super-resolution in settings where paired LR–HR training data is unavailable. Several reviewers find the idea of mining unpaired LR and HR patches from a single image using depth variations to be interesting, and they also appreciate the attempt to leverage vision–language models such as CLIP to help bridge the gap between low- and high-resolution imagery.

However, several concerns are raised and appear repeatedly across the reviews. First, reviewers PnCK and r1tK point out that treating depth as a direct proxy for image quality may be an oversimplification. In real images, quality is often affected by multiple factors simultaneously, including motion blur, depth of field, texture complexity, and sensor noise, which are not necessarily correlated with distance.

Another recurring concern comes from reviewers RjqZ and 89QH, who question whether high-level language representations can really preserve fine-grained visual content. This is especially unclear for patches that contain mostly textures, flat color regions, or only partial semantics. In these cases, it is not clear that linguistic signals alone are sufficient to prevent semantic ambiguity or content mismatch.

Finally, reviewers RjqZ, r1tK, and 89QH note that the experimental evaluation relies primarily on older GAN-based super-resolution methods. The lack of comparisons with more recent diffusion-based or modern blind and unpaired SR approaches makes it difficult to assess the competitiveness of the proposed method relative to the current state of the art.

**Reviewer Concerns:**

The authors do not provide a rebuttal or substantive response to the reviews. As a result, the concerns raised by the reviewers are not directly addressed.

**Reviewer Scores:**

Given the lack of a rebuttal, none of the reviewers’ scores is expected to change.

---

### Decision · Program_Chairs · 2026-01-26

Reject